# Association of Hypomagnesemia and Liver Injury, Role of Gut-Barrier Dysfunction and Inflammation: Efficacy of Abstinence, and 2-Week Medical Management in Alcohol Use Disorder Patients

**DOI:** 10.3390/ijms231911332

**Published:** 2022-09-26

**Authors:** Evan J. Winrich, Khushboo S. Gala, Abhas Rajhans, Christian D. Rios-Perez, Amor J. Royer, Zarlakhta Zamani, Ranganathan Parthasarathy, Luis S. Marsano-Obando, Ashutosh J. Barve, Melanie L. Schwandt, Vatsalya Vatsalya

**Affiliations:** 1Department of Medicine, University of Louisville, Louisville, KY 40202, USA; 2Alcohol Research Center, University of Louisville, Louisville, KY 40202, USA; 3Clinical Laboratory for the Intervention Development of AUD and Organ Severity, University of Louisville, Louisville, KY 40202, USA; 4Department of Neuroscience, University of California-Los Angeles, Los Angeles, CA 90095, USA; 5Robley Rex Louisville VA Medical Center, Louisville, KY 40206, USA; 6National Institute on Alcohol Abuse and Alcoholism, Bethesda, MD 20892, USA

**Keywords:** alcoholic liver disease, alcohol use disorder, early-stage ALD, heavy drinking, hypomagnesemia

## Abstract

(1) We investigated the involvement of serum magnesium level in early alcoholic liver disease (ALD), gut barrier dysfunction, and inflammation in alcohol use disorder (AUD) patients; and lastly, the efficacy of 2-week abstinence and medical management to alleviate hypomagnesemia. (2) Forty-eight heavy drinking AUD patients (34 males (M)/14 females (F)) participated in this study. Patients were grouped by serum alanine aminotransferase (ALT) level (a marker of liver injury) as group 1 (Group 1 (Gr.1); ALT ≤ 40 U/L, 7M/8F, without any indication of early-stage ALD) and group 2 (Group 2 (Gr.2); ALT > 40 U/L, 27M/6F or early-stage ALD). These patients were sub-divided within each group into patients with normal magnesium (0.85 and more mmol/L) and deficient magnesium (less than 0.85 mmol/L) levels. All participants were assessed at baseline (BL) and received standard medical management for 2 weeks with reassessment at the treatment end (2w). (3) Female participants of this study showed a significantly lower baseline level of magnesium than their male counterparts. Gr.2 patients showed a greater propensity in the necrotic type of liver cell death, who reported higher chronic and recent heavy drinking. Magnesium level improved to the normal range in Gr.2 post-treatment, especially in the hypomagnesemia sub-group (0.77 ± 0.06 mmol/L (BL) vs. 0.85 ± 0.05 mmol/L (2w), *p* = 0.02). In Gr.2, both apoptotic (K18M30) and necrotic (K18M65) responses were significantly and independently associated with inflammasome activity comprising of LBP (Lipopolysaccharide binding-protein) and TNFα (Tumor necrosis factor -α), along with serum magnesium. (4) In AUD patients with liver injury, 2-week medical management seems to improve magnesium to a normal level. This group exhibited inflammatory activity (LBP and TNFα) contributing to clinically significant hypomagnesemia. In this group, the level of magnesium, along with the unique inflammatory activity, seems to significantly predict apoptotic and necrotic types of hepatocyte death.

## 1. Introduction

Alcohol-associated liver disease (ALD) is a spectrum of liver diseases, including steatosis, hepatitis, and fibrosis/cirrhosis, caused secondary to chronic excessive alcohol intake [1]. Several clinical studies investigate patients diagnosed with the advanced form of ALD; albeit those with early ALD are often missed [2,3]. This has led to a significant gap in our knowledge about the pathophysiology of early ALD. The detection of specific biomarkers that could characterize early ALD would help to serve as diagnostic tools for the better management of these patients, and offer an assessment of the treatment course [4].

Alcohol intake leads to various electrolyte disturbances, including deficiencies in magnesium [5]. Hypomagnesemia has been observed with both acute as well as chronic alcohol abuse [6,7,8,9]. Research has shown that chronic alcohol use alters magnesium homeostasis and its transport in liver cells [6]. One study has even shown that increasing hypomagnesemia is found on muscle biopsy with the increasing severity of liver injury [10]. Another recent study showed that heavy and chronic alcohol drinking is involved in hypomagnesemia that could be contributory to liver injury in early-stage ALD [11].

There is a lack of data on electrolyte imbalances in early-stage alcohol-associated liver disease. This is a field of continued clinical interest, as evidenced by recent papers that showed that mortality is lower in patients with liver disease who took high doses of magnesium [12]. There is ongoing research to explore the role of electrolyte imbalances as biomarkers for ALD, and the pathophysiology of these disturbances [13]. Alcohol could dysregulate the immunological status that eventually adversely impacts upon liver health and brain activity [14]. Importantly, we do not know if the pattern of alcohol drinking is involved in the type and extent of liver cell death. The liver cell death markers K18M65 and K18M30 represent the necrotic and apoptotic types of liver cell death [3,15]. The shift in the type of liver cell death may show early changes along with the liver injury and/or heavy drinking patterns, though such characterization has not been investigated yet. Additionally, these alterations may have variability in the degree and severity by the modifying effects of age and sex of the individuals involved in heavy and chronic drinking [16].

In this study, we aim to characterize the level of heavy drinking markers by the extent of the liver cell death marker, K18M65, in alcohol use disorder (AUD) patients. We further investigate the role of serum magnesium in context of heavy drinking pattens consequential in liver injury/liver cell death, cytokine and pro-inflammatory responses, and candidate pathological pathways involved in liver injury at baseline and post-2-week standard of care (SOC, 2w) assessment. Lastly, we explored the role of modifiers of ALD, such as age and sex variables/factors.

## 2. Results

### 2.1. Demographics, Drinking, and Nutritional Status

Twenty-five out of 48 AUD patients reported hypomagnesemia at admission. There was no difference in age and BMI between the two groups (grouped by the level of ALT as the primary factor). In total, 17 out of 33 (51%) AUD patients with mild liver injury at admission (Gr.2) exhibited hypomagnesemia, whereas 8 out of 15 (53%) AUD patients without liver injury at admission (Group 1 or Gr.1) were found to have hypomagnesemia. Out of the 15 Gr.1 patients, the number of female patients who exhibited hypomagnesemia exceeded their male counterparts by three-fold (Table 1). In Group 2 (Gr.2), there were notably more males than females (4.5 times more), and males who exhibited hypomagnesemia outnumbered their female counterparts by more than three-fold. Lifetime drinking (by the number of years) was significantly higher in Gr.2 compared to Gr.1. HDD90 and NDD90 heavy drinking markers were numerically higher in Gr.2 as well.

### 2.2. Baseline Magnesium Level and Nutritional Status

The serum magnesium level was deficient in 8/15 patients of Gr.1, and 17/33 patients of the Gr.2 (Table 1). Notably, both the groups did not show any statistical or numerical difference in the CONUT scores, thus excluding the dietary deficiency could be justified. Gr.2 patients with hypomagnesemia had elevated CONUT scores compared to the Gr.2 patients with a normal magnesium level (Table 1). However, the hypomagnesemia exhibited in the patients of both the groups had similarly deficient levels of magnesium (Figure 1a). When we reviewed the SM between the sexes in the whole subject pool, we found that the females had a greater adverse response to heavy drinking, which corresponded to a lower magnesium level (Figure 1b). There was no difference in the serum magnesium level between the males and females within each group (Figure 1c).

### 2.3. Baseline Gut-Dysfunction and Pro-Inflammatory Response

LPS, LBP, and +sCD14 were numerically higher in Gr.2 compared to Gr.1 values. TNF-α was a distinct pro-inflammatory cytokine that showed significant elevation only in Gr.2. In Gr.2, IL-6 and MCP-1 were numerically high, and IL-8 and IL-1β were lower, suggesting immunological uniqueness by the staging of the liver disease (Table 1).

### 2.4. Baseline Liver Injury, and Liver Cell Death

As anticipated, liver injury characterized by ALT (as well as AST) was significantly higher in Gr.2 by approximately four-fold, and both were clinically and numerically significant compared to the Gr.1 values (Table 1). Liver injury progression as recorded by the AST:ALT ratio was comparable in both the groups by their individual numerical value and was not statistically different. Both the necrotic marker, K18M65 and the necrosis trend, K18M65:M30 were markedly and significantly elevated in the Gr.2. Importantly, Gr.2 patients who exhibited hypomagnesemia had approximately double the level of necrosis and necrotic trend compared to the Gr.1 patients exhibiting hypomagnesemia.

The markers of heavy alcohol drinking in the statistical paradigm grouped by normal or high levels of K18M65 showed significant elevations in candidate recent heavy and chronic drinking markers. HDD90 and NDD90 as the acute and chronic markers of heavy drinking, respectively, and LTDH, were all significantly high in patients who also had reported clinically significant K18M65 levels (Figure 2a–c). Such differences were not evident with the differences in the K18M30 levels.

### 2.5. Association of Magnesium with Gut-Dysfunction, Inflammation, and Liver Cell Death Markers

Identifying the differences in the heavy drinking markers that go along with the liver cell death in the previous section, we further investigated the liver cell death response for its association in AUD patients without (Gr.1) and with liver injury (Gr.2). K18M65:M30 (or the ratio of K18M65 by K18M30), a necrotic trend marker, showed a positive corresponding increase response with the recent heavy drinking markers, NDD90 (Figure 3a) and HDD90 (Figure 3b). However, this response was significant only in context of serum magnesium as a co-independent variable, along with recent heavy drinking markers (independently, NDD90: R^2^ = 0.253 at *p* = 0.012; HDD90: R^2^ = 0.219 at *p* = 0.024) in this multivariable regression model. We did not find any association in magnesium and liver cell death markers when there was no liver injury in the Gr.1 cohort. We also did not find any such response with K18M65 or K18M30 independently in the same statistical model.

A significant effect in the association of baseline serum magnesium and liposaccharide (LPS) was found at *p* = 0.042 (adjusted R^2^ = 0.227) that augmented to *p* = 0.021 (adjusted R^2^ = 0.413) with TNF-α as an additional covariate only in Gr.1, showing the pre-injury arrangement of inflammasome activity. However, in Gr.2, the response from both the apoptotic (K18M30) (*p* = 0.003 at R^2^ = 0.428) and necrotic (K18M65) (*p* = 0.032 at R^2^ = 0.302) markers, respectively, exhibited a significant association with the inflammasome activity comprising LBP and TNF-α levels, along with a lower level of serum magnesium (Figure 4a,b). This relationship was resolved at 2w (Figure 4c,d) by the alleviation in magnesium level. It is important to note that the change in magnesium was crucial for the drop in K18M65 and K18M30 levels, post-treatment.

### 2.6. Post-SOC Evaluation

Normal magnesium level was attained in Gr.2 with two weeks of medical management, importantly in the hypomagnesemia sub-group. In Gr.1, this change was not observed in the patients with baseline hypomagnesemia. Performing repeated analyses of variance for the baseline to the 2w K18M65:M30 ratio, K18M65, and K18M30 (markers of cell death) among the hypomagnesemia sub-groups only for both the groups, we found that the drop in the Gr.2 patients with hypomagnesemia was larger and exhibited a statistically significant main effect for the K18M65:M30 (*p* = 0.022) and K18M30 ratio (*p* = 0.051). To confirm the true positivity of this result, we performed Receiver Operating Characteristic (ROC) curve analysis for the restoration of magnesium level at post-treatment stage in Gr.2. The area under the curve (AUROC) was 0.857 (a very good fit for true positivity) at high significance, *p* = 0.014, compared to their corresponding baseline values (Figure 5). Notably, the 2w TNF-α was correspondingly elevated, along with K18M65 in the Gr.2 patients, a potential explanation of the continuity of the necrotic process and with K18M65 remaining at a sub-clinical high level at 2w, especially in the hypomagnesemia sub-group (Table 2).

## 3. Discussion

More than half of the AUD patients admitted in the study, regardless of the study randomization, exhibited hypomagnesemia. Most of the study patients exhibited hypomagnesemia; those who exhibited hypomagnesemia with liver injury also reported clinically and statistically significant elevation in the AST levels. Thus, all AUD patients regardless of liver injury could potentially have low magnesium, though the progression of liver injury could have a higher probability in patients with AUD and hypomagnesemia. Hypomagnesemia can be caused by one of three pathophysiologic mechanisms, including reduced intestinal absorption, increased urinary losses, or intracellular shift [17]. It is interesting to note that heavy and chronic alcohol drinking could cause hypomagnesemia via all three mechanisms. Both heavy drinking episodes and their frequency could contribute to the progressive deficiency of magnesium. Notably, females were found to have a greater adverse response to heavy drinking that corresponded to the lower magnesium level than their male counterparts, suggesting sexual dimorphism in the context of pathological consequence in the magnesium level from heavy and chronic drinking. In this study, the difference in the heavy drinking markers with the necrotic cell death of the hepatocytes was established, which was a novel finding. We found that the hypomagnesemia was evident in the AUD patients with normal liver enzymes, who also showed exacerbated gut dysfunction uniquely characterized by more than 3-fold higher values of LBP. The ratio of K18M65 by K18M30 is an indication of necrotic shift, and we found that this shift positively corresponded to the recent heavy alcohol drinking that attained statistical significance in the context of the magnesium level. This is an interesting finding, since this ratio has been also observed in AUD patients with early-stage ALD, as well as in alcoholic hepatitis (AH) patients, in which it indicates the necrotic shift that corresponds with the severity of liver status [18,19].

Contributing factors leading to hypomagnesemia include a poor nutritional status leading to decreased absorption [6,20,21], increased urinary losses [20,22], and impaired magnesium homeostasis [9]. Overt hypomagnesemia leads to weakness, ataxia, cramps, tetany, seizures, and arrhythmias/electrocardiographic changes [23]. Newer research has been looking at different stages/level of magnesium deficiency and has found that even subclinical magnesium deficiency may be one of the leading causes of chronic diseases and early mortality [24]. Clinical implications of hypomagnesemia in liver disease are significant and may lead to supplementation being one of the recommended modalities in the medical management of alcohol-associated liver disease.

Studies in alcohol-fed rats have shown that magnesium supplementation helped with oxidative stress and tissue damage, as evidenced by elevated total antioxidant status (TAS) in serum, the activity of glutathione peroxidase, and the ratio of reduced glutathione to oxidized glutathione (GSH/GSSG) in the liver, as well as tissue histopathological changes [25]. Magnesium supplementation in rats with liver damage has been shown to have anti-fibrotic properties, with improvement in oxidant and antioxidant parameters and histopathological examination [26].

The 2w SOC assessment yielded some interesting findings. Two weeks of SOC sufficiently treated the deficiency in the magnesium levels of AUD patients who exhibited baseline liver injury; this corresponded well with the lowering of the AST and AST:ALT levels in these AUD patients [27]. ALT level was higher in Gr.2 after 2 weeks of the treatment; this could be attributed to the persisting inflammation that could not be alleviated by the end of the study. This could indicate that the active inflammation may take more time to alleviate than the ongoing cell death and ALD progression. One trial of magnesium supplementation study on human chronic alcohol users in Finland showed that when given magnesium supplementation, patients have a lowering of serum AST level and may have a decreased risk of death from alcoholic liver disease [28]. Due to its role in cellular regeneration and anti-oxidation, hypomagnesemia in the early stages of liver disease may be one of the factors leading to the progression of this liver disease. ALT also remained borderline high at 2w SOC assessment; however, it was corrected significantly from the baseline values. This drop could be validated by the response change in the necrosis marker, K18M65, which correspondingly dropped to the borderline high level. Furthermore, the K18M30 values normalized completely at 2w in the same cohort. Studies in non-alcoholic fatty liver disease support the protective role of magnesium as a disease control [29].

All of these patients were inpatient, and they maintained abstinence, and active alcohol drinking was not a contributing factor. Thus, patients receiving SOC with nutritional therapy and the management of any of the clinical presentation of AUD, including withdrawal, were apt for recovery [30] in this patient cohort with mild liver injury exhibiting hypomagnesemia. None of these patients had alcohol-associated cirrhosis or hepatitis. However, one study showed that compared to patients with non-alcoholic steatosis, patients with steatohepatitis had a lowered serum magnesium level [31].

This study had several limitations due to the extent and scope of the study design. This study was a proof-of-concept study, and the sample size was relatively small. This limited the ability of the paradigm to evaluate the vulnerability in females and males independently. Females were less in number in each group by the sub-group categories; thus, running statistics was not feasible at the sub-group level (specially within sub-group male to female comparisons). A smaller sample size also poses a greater challenge for research compliance, since the post-treatment has lesser subjects (compared to the baseline strength) who provided the research samples. This clinical study was designed as a cross-sectional longitudinal treatment by time investigation. However, the study aims were secondary under a larger protocol; thus, studying all the corresponding biomarkers were not within the scope at the present time when we completed the investigation. The study was aimed at 2 weeks of treatment, given that the anticipated mild liver injury should be treated within the timeframe effectively. It was a revelation that many patients will progress regardless of their ALT or AST levels. Thus, it is important to look for the mechanisms that could predict the underlying course of liver health and pathology. We did not test magnesium levels in AUD patients with cirrhosis (AC) and hepatitis (AH); thus, we do not know what unique differences in the response change could have progressed along with the advancement of the ALD. We have started a new protocol to study magnesium response in AH patients. Studies show that advanced ALD and the level of magnesium could be a clinical direction of investigation [32]. A necrosis and apoptotic assessment could reveal the pathological status better; however, such assays are not readily available as point of care. The magnesium level could show a corresponding change in the underlying necrotic and apoptotic cascade, which is a readily available point of care (POC) and could be helpful in suggesting the medical management of the AUD patients who could be at high risk of developing and progressing into an advanced form of ALD.

Further hypothesis-driven well-structured treatment trials and large longitudinal studies are warranted to explore the potential of magnesium as an adjunct therapy of liver disease. Our findings supported the role of serum magnesium as a potential biomarker for alcohol-associated early-stage liver injury and liver cell death. Our study also describes the corresponding changes in the magnesium level, along with liver cell death and its alleviation with the treatment course over time.

## 4. Materials and Methods

This investigation is a secondary aim of a larger clinical investigation (NCT#00106106) that was conducted at the National Institute on Alcohol Abuse and Alcoholism (NIAAA) at the National Institutes of Health (NIH), Bethesda, MD. A total of 48 male and female AUD patients within an age range of 21–65 yrs participated in this study, receiving a detox program for one month at the NIH Clinical Center. All study patients were diagnosed with AUD based on DSM-IV, TR edition [33]. The alcohol dependence module of the Structured Clinical Interview I, and alcohol withdrawal were administered for reaching the AUD diagnosis. All study patients received medical management and addiction therapy throughout their stay; more details are available in our previous publications [34,35]. Further detailed information on admission, exclusion, and inclusion could be reviewed in a primary publication on investigational drug efficacy [36].

A few important exclusion criteria are described here: (i) the presence of severe psychiatric and/or somatic illnesses, including advanced lung disease, unstable cardiovascular disease (decompensation, as demonstrated through chest X-ray and pathological electrocardiogram), and/or renal failure (creatinine clearance < 30 mL/min). Other exclusion criteria were (ii) the presence of HIV, (iii) pregnancy or ongoing breastfeeding, and (4) pronounced anxiety provoked by enclosed spaces, and/or positive urine screen for any illicit drug. No AUD patient exhibited any clinical evidence of advanced ALD or gout disease. All patients received standard clinical inpatient care for alcohol detoxification and medical management according to the “Human Subjects Protection” guidelines of NIH.

### 4.1. Demographics, Drinking, and Laboratory Evaluations

Blood was drawn once patients consented to participate in the inpatient study. On admission, blood samples were collected for a serum chemistry panel (Table 1) that included tests for liver injury and the serum magnesium (SM) level. Demographic information (age, sex, and body mass index (BMI)) and drinking history were also collected for the study. Heavy drinking measures were collected from the Timeline Follow-back questionnaire [37]. Markers of heavy drinking derived from the TLFB reported in the past 90 days were “Total Drinks” (TD90), “Number of Drinking Days” (NDD90), “Number of Non-Drinking Days” (NNDD90), “Average Drinking per Drinking Days” (AvgDPD90), and “Heavy Drinking Days” (HDD90). Chronic drinking was reported using lifetime drinking history (LTDH) [38]. We used the “Controlling Nutritional Status Test” (CONUT) information on these patients to assess their nutritional status [39]. The alanine aminotransaminase (ALT) level was used as a biomarker for early liver injury (Medline Plus-National Institutes of Health, 2014). Normal serum alanine aminotransferase (ALT) values were set at <40 IU/L, and patients were categorized as Group 1: those with normal ALT level; and Group 2: those with ALT > 40 IU/L, as indicative of mild liver injury. The reference normal range for serum magnesium is 0.85–1.10 mmol/L. Patients with serum magnesium <0.85 mmol/L were considered as having magnesium deficiency. All laboratory assays were performed by the Department of Laboratory Medicine at NIH Bethesda MD, per its guidelines (https://medlineplus.gov/ency/article/003487.htm 9 January /2022).

### 4.2. Laboratory Assays

Frozen plasma samples at −80 degree Celsius were thawed and assayed. Plasma cytokeratin 18 whole protein (K18M65) and caspase-cleaved fragment (K18M30) were analyzed using enzyme-linked immunosorbent assay (ELISA) (Peviva-VLVbio, Nacka, Sweden) according to the manufacturer’s instructions. Clinically significant K18 is as follows: K18M65 > 500 U/L or K18M30 > 250 U/L (was used for categorical differences in the Figure 2 analyses). Plasma pro-inflammatory cytokines, TNF-α, interleukin 1β, interleukin 6, and interleukin 8 (IL-1β, IL-6, and IL-8), PAI-1, and monocyte chemoattractant protein-1 (MCP-1) were obtained via multianalyte chemiluminescent detection using Mulliplex kits (Millipore, Billerica, MA) on the Luminex (Luminex, Austin, TX, USA) platform according to manufacturers’ instructions. Plasma lipopolysaccharide (LPS) and lipopolysaccharide (LBP) levels were assayed using the Kinetic Chromogenic Limulus Amebocyte Lysate Assay (Lonza, Walkersville, MD) according to the manufacturer’s instructions.

### 4.3. Statistical Analysis

One-way ANOVA was used to evaluate demographic and drinking history measures. Univariate analysis of covariance (ANCOVA) was used to evaluate differences in the serum magnesium level in both the groups and by the modifiers of ALD, primarily by sex, within each of the liver injury groups as factors. Drinking history and other demographic factors were tested as confounders (covariates) of the extent and progression of liver injury. Linear regression analysis was used to characterize the association of liver injury markers and SM independently (or with covariables in the context of drinking history measures, sex, cytokines, and gut permeability factors). Repeated analyses of variance were performed to evaluate the treatment effects of the detox program and intervention on restoring normal SM by group over time (at 2w). To eliminate the possibility of a type I error, Receiver Operating Characteristic (ROC) analysis and area under the ROC (AUROC) were used to estimate the probability of the outcome of treatment in Gr.2 patients with documented hypomagnesemia, compared to those without, at the end of the study. SPSS 27.0 (IBM Chicago, IL, USA) and Microsoft 365 Excel (MS Corp, Redmond WA) were used for statistical analysis and data computation. Statistical significance was established at *p* < 0.05. Data are expressed as M ± SD (Mean ± standard deviation), unless otherwise noted.

## Figures and Tables

**Figure 1 ijms-23-11332-f001:**
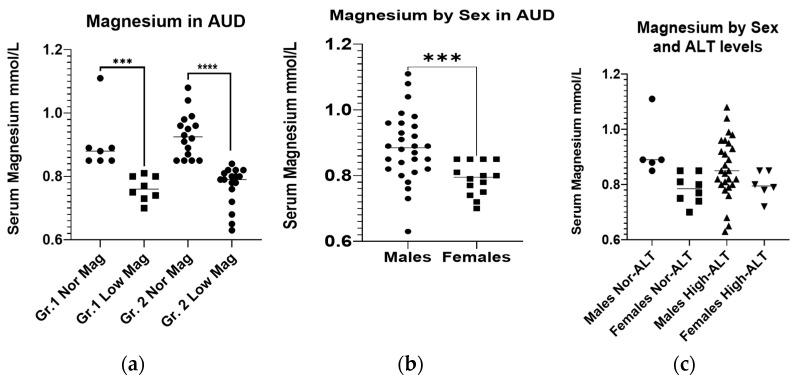
Initial serum magnesium levels in AUD patients. (**a**): Initial serum magnesium levels in AUD patients without (Gr.1) and with liver injury (Gr.2), sub-grouped by the factor of magnesium. (**b**): Initial serum magnesium levels in male and female AUD patients. (**c**): Initial serum magnesium levels in male and female AUD patients without (Gr.1) and with liver injury (Gr.2). The normal reference range for serum magnesium is 0.85–1.10 mmol/L. Data presented as Mean ± Standard Deviation. Statistical significance was set at *p* < 0.05. *** *p* < 0.001. **** *p* < 0.0001.

**Figure 2 ijms-23-11332-f002:**
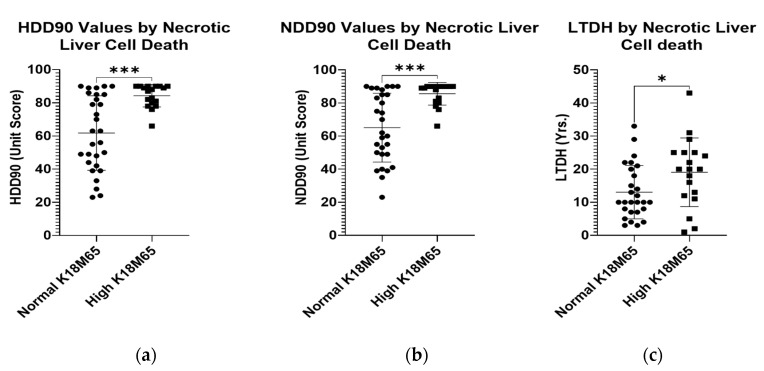
Differences in chronic and acute heavy drinking markers in patients with normal and high K18M65 levels. (**a**): Difference in the mean HDD90 (heavy drinking days past 90 days) values in patients with AUD exhibiting clinically significant K18M65 levels compared to those with normal K18M65. (**b**): Difference in the mean NDD90 (number of drinking days past 90 days) values in patients with AUD exhibiting clinically significant K18M65 levels compared to those with normal K18M65. (**c**): Difference in the chronic alcohol drinking marker, LTDH (Lifetime drinking history (in years)) in patients with AUD exhibiting clinically significant K18M65 levels compared to those with normal K18M65. Data presented as Mean with Standard Deviation (M ± SD). Statistical significance was set at *p* < 0.05. * *p* < 0.05, *** *p* < 0.001.

**Figure 3 ijms-23-11332-f003:**
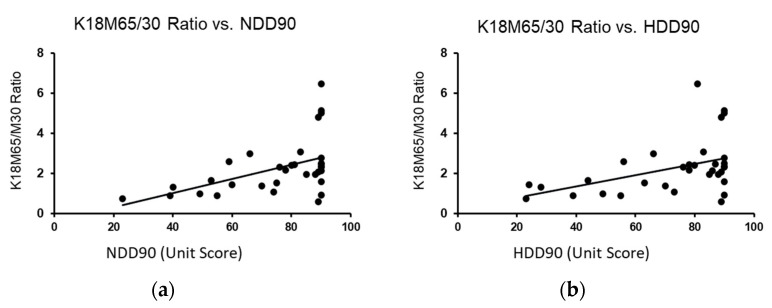
Association of recent heavy drinking marker with the liver necrotic ratio in AUD patients with liver injury (Gr.2) with the co-independent variable being magnesium. (**a**): Association of NDD90 and K18M65/M30 ratios in all AUD patients with liver injury (Gr.2). (**b**): Association of HDD90 and K18M65/M30 ratios AUD patients of Gr.2. Statistical significance was set at *p* < 0.05.

**Figure 4 ijms-23-11332-f004:**
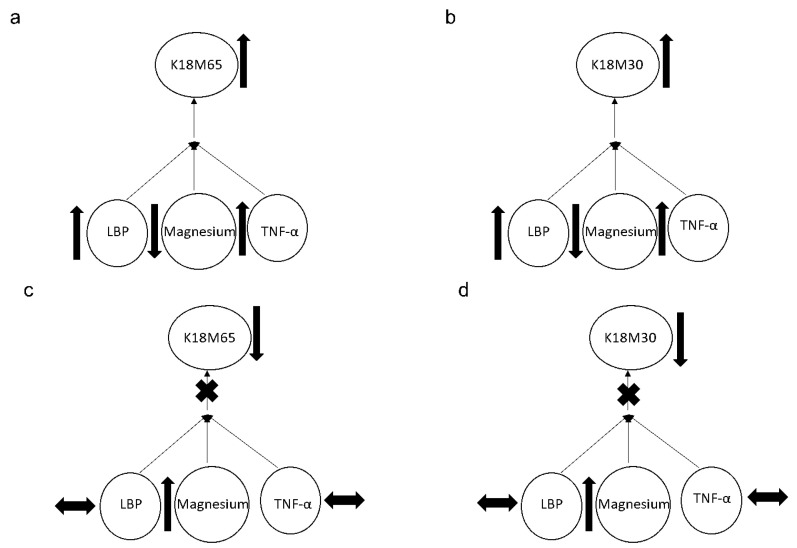
Association of the apoptotic (K18M30) and necrotic (K18M65) markers with magnesium and inflammasome activities represented by the LBP and TNF-α levels. (**a**,**b**): Association of the apoptotic (K18M30) and necrotic (K18M65) markers with magnesium and inflammasome activities represented by LBP and TNF-α levels at the baseline of the trial. (**c**,**d**): Association of the apoptotic (K18M30) and necrotic (K18M65) markers with magnesium and inflammasome activities represented by LBP and TNF-α levels at 2 weeks of trial. Statistical significance was set at *p* < 0.05. Arrow up: elevation in values. Arrow down: lowering in values. Arrow sideways: minimal change.

**Figure 5 ijms-23-11332-f005:**
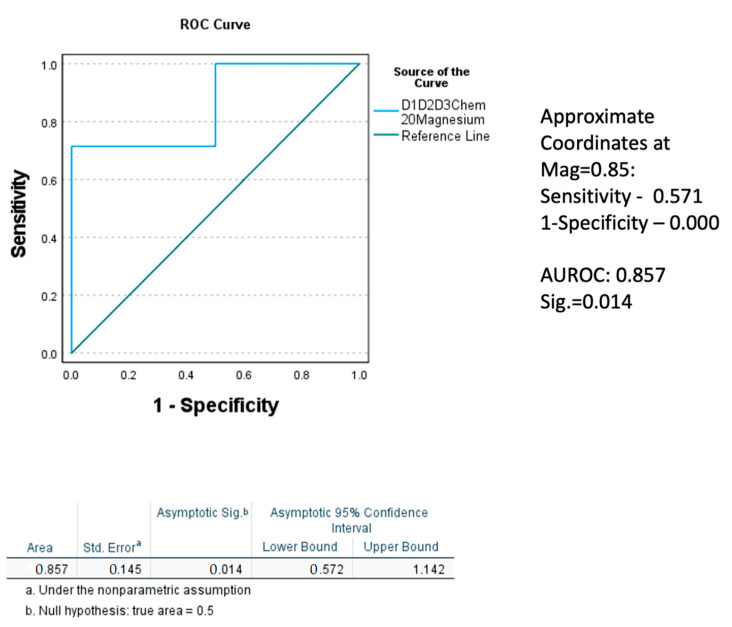
Receiver operating characteristic (ROC) curve demonstrating the true positivity of the level of effect in the replenishment of magnesium level at 2 weeks, factored by the level of magnesium at admission in Gr.2. Statistical significance was set at *p* < 0.05.

**Table 1 ijms-23-11332-t001:** Baseline assessment of demographics, drinking history, liver injury, magnesium and nutritional level, blood cell measures, cytokine and gut-permeability markers, and liver cell death markers.

Measures	Group 1 (Normal Initial ALT, Gr.1)			Group 2 (Elevated Initial ALT, Gr.2)			*Between Group* *p*-Value
	Normal Mg (n = 7; 46.67%)	Low Mg (n = 8; 53.33%)	Total (n = 15; 31.25%)	Normal Mg (n = 16; 48.48%)	Low Mg (n = 17; 51.52%)	Total (n = 33; 68.75%)	
Age (years)	37.64 ± 10.5	42.9 ± 12.4	40.45 ± 11.4	41.4 ± 9.5	47.3 ± 9.5	44.45 ± 9.8	NS
BMI (kg/m^2^)	31.31 ± 8.2	25.9 ± 7.4	28.45 ± 8.0	25.9 ± 4.7	26.1 ± 2.9	25.99 ± 3.7	NS
Sex (F/M)	2/5	6/2	8/7	2/14	4/13	6/27	NA
Baseline Drinking History							
TD90	1511.2 ± 653.3	809.2 ± 598.0	1160.21 ± 703.33	1111.2 ± 593.0	1048.65 ± 440.7	1078.98 ± 512.80	NS
HDD90 ^c^	75.4 ± 14.0	54.57 ± 21.9	65.00 ± 20.71	68.94 ± 22.2	76.24 ± 21.9	72.69 ± 21.18	NS
AvgDPD90	19.36 ± 6.54	14.19 ± 7.6	16.78 ± 7.33	15.46 ± 6.5	13.18 ± 4.5	14.29 ± 5.60	NS
NDD90 ^c^	76.86 ± 56.86	56.86 ± 23.25	66.86 ± 21.48	72.00 ± 20.71	78.49 ± 15.85	75.49 ± 18.40	NS
LTDH ^c^	12.5 ± 6.7	9.63 ± 5.3	10.86 ± 5.86	17.3 ± 10.1	17.8 ± 10.4	17.56 ± 10.06	0.025
Baseline Liver Injury Markers							
ALT (IU/L) ^c,d^	27.86 ± 7.0	25.25 ± 10.5	26.47 ± 8.86	87.19 ± 48.13	109.24 ± 62.8	98.55 ± 56.43	NA
AST (IU/L) ^c,d^	31.14 ± 9.8	37.50 ± 26.5	34.53 ± 20.09	115.56 ± 95.5	144.94 ± 105.8	130.70 ± 100.45	<0.01
AST: ALT	1.16 ± 0.4	1.43 ± 0.6	1.30 ± 0.51	1.23 ± 0.6	1.35 ± 0.8	1.29 ± 0.70	NS
Baseline Magnesium Levels							
Serum Mg mmol/L ^a,b^	0.9 ± 0.09	0.76 ± 0.04	0.83 ± 0.10	0.93 ± 0.07	0.77 ± 0.06	0.85 ± 0.10	NS
Baseline Nutritional Status							
CONUT	1.00 ± 1.0	1.00 ± 1.7	1.00 ± 1.36	0.75 ± 0.9	1.18 ± 1.1	0.97 ± 1.05	NS
Baseline Blood Cell Types							
WBC (K/uL) ^c^	7.03 ± 3.2	7.93 ± 3.05	7.51 ± 3.03	6.42 ± 2.3	5.65 ± 2.1	6.03 ± 2.19	0.061
AMC (K/uL)	0.62 ± 0.26	0.46 ± 0.21	0.53 ± 0.24	0.58 ± 0.28	0.46 ± 0.16	0.52 ± 0.23	NS
ANC (K/uL)	4.29 ± 2.4	4.66 ± 2.5	4.49 ± 2.37	3.64 ± 1.7	3.49 ± 1.7	3.56 ± 1.68	NS
Baseline Candidate Cytokine Response							
IL-1β (pg/mL) ^a^	0.66 ± 0.50	0.57 ± 0.60	0.62 ± 0.53	0.50 ± 0.29	0.45 ± 0.23	0.47 ± 0.26	NS
IL-6 (pg/mL) ^a,c,d^	2.25 ± 1.15	3.68 ± 4.41	2.97 ± 3.18	3.64 ± 2.16	3.98 ± 3.94	3.82 ± 3.19	NS
TNF-α (pg/mL)	1.74 ± 0.77	1.17 ± 0.52	1.45 ± 0.70	1.93 ± 0.59	2.26 ± 1.15	2.11 ± 0.93	0.025
IL-8 (pg/mL) ^a^	13.25 ± 26.19	4.63 ± 5.23	8.94 ± 18.68	4.05 ± 2.09	9.00 ± 13.92	6.69 ± 10.42	NS
MCP-1 (pg/mL) ^a^	96.03 ± 29.75	110.87 ± 67.01	103.45 ± 50.41	115.91 ± 56.43	115.66 ± 72.96	115.78 ± 64.66	NS
Baseline Candidate Gut-dysfunction Markers							
LPS (EU/mL)	0.078 ± 0.06	0.080 ± 0.05	0.08 0.05	0.106 ± 0.06	0.110 ± 0.06	0.11 ± 0.06	NS
LBP (ng/mL)	624.51 ± 742.92	2009.87 ± 3374.97	1317.19 ± 2455.31	2497.59 ± 3096.29	1759.19 ± 2750.63	2092.66 ± 2886.01	NS
+sCD14 (×10 pg/mL)	8865.95 ± 2238.92	8962.71 ± 1509.51	8917.56 ± 1813.29	9193.12 ± 1997.19	9744.99 ± 1614.74	9477.42 ± 1803.29	NS
Baseline Liver Cell Death Markers							
K18M65 (IU/L)	138.62 ± 63.85	456.42 ± 528.24	308.11 ± 410.12	856.21 ± 1083.77	922.37 ± 827.66	890.29 ± 945.63	0.027
K18M30 (IU/L)	514.08 ± 854.73	278.89 ± 165.09	388.65 ± 584.36	361.13 ± 415.01	378.20 ± 342.46	378.20 ± 342.46	NS
M65:M30	0.682 ± 0.59	1.333 ± 0.78	1.03 ± 0.76	2.325 ± 1.50	2.214 ± 1.26	2.27 ± 1.36	0.002

BMI: Body mass index, TD90: Total drinks past 90 days, HDD90: heavy drinking days past 90 days, AvgDPD90: Average drinks per drinking day past 90 days, NDD90: number of drinking days past 90 days, NNDD90: number of non-drinking days past 90 days, LTDH: lifetime drinking history (in years), ALT: serum alanine aminotransferase, AST: serum aspartate aminotransferase, AST:ALT—ratio of AST by ALT, CONUT: Controlling Nutritional Status Test (unit: numerical), WBC: white blood cells count, AMC: absolute monocyte count, ANC: absolute neutrophil count, Il1β: interleukin 1 beta, IL-6: interleukin 6, TNFα: tumor-like necrotic factor alpha, LPS: lipopolysaccharide, LBP: LPS binding protein, sCD14: soluble cell of differentiation type 14, K18M65: soluble CK18, K18M30: caspase-cleaved fragment of CK18, M65:M30—ratio of K18M65 by K18M30. ^a^ Statistically significant difference between the hypomagnesemia sub-groups of the two groups. ^b^ Statistically significant difference between the sub-groups exhibiting normal magnesium levels of the two groups. ^c^ Statistically significant difference between the sub-groups of Gr.1. ^d^ Statistically significant difference between the sub-groups of Gr.2.

**Table 2 ijms-23-11332-t002:** Post-study assessment of liver injury, magnesium level, blood measures, cytokine, and gut-permeability markers and liver cell death markers.

Measures	Group 1 (Normal Initial ALT, Gr.1)			Group 2 (Elevated Initial ALT, Gr.2)			*Between Group* *p*-Value
	Normal Mg	Low Mg	Total	Normal Mg	Low Mg	Total	
Post-Study Liver Injury Markers							
ALT (IU/L)	58.33 ± 34.7	50.67 ± 36.7	54.50 ± 32.24	68.75 ± 18.4	56.00 ± 22.3	61.67 ± 20.50	NS
AST (IU/L)	78.0 ± 79.98	68.67 ± 58.53	73.33 ± 62.89	36.00 ± 6.98	30.40 ± 7.09	32.89 ± 7.22	0.073
AST: ALT	1.16 ± 0.56	1.27 ± 0.18	1.22 ± 0.38	0.54 ± 0.13	0.61 ± 0.28	0.58 ± 0.22	0.001
Post-Study Magnesium Levels							
Serum Mg mmol/L	0.90 ± 0.06	0.80 ± 0.13	0.85 ± 0.10	0.92 ± 0.07	0.85 ± 0.05	0.88 ± 0.07	NS
Post-Study Candidate Cytokine Response							
IL-1β (pg/mL)	0.64 ± 0.39	0.62 ± 0.93	0.63 ± 0.68	0.38 ± 0.25	0.78 ± 0.96	0.59 ± 0.74	NS
IL-6 (pg/mL)	2.44 ± 1.02	3.44 ± 3.56	2.94 ± 2.57	3.30 ± 2.29	2.95 ± 0.95	3.11 ± 1.69	NS
TNF-α (pg/mL) ^a^	2.03 ± 0.68	1.49 ± 0.64	1.76 ± 0.69	2.45 ± 0.85	2.61 ± 0.78	2.53 ± 0.85	0.004
IL-8 (pg/mL)	2.36 ± 0.47	4.35 ± 6.02	3.36 ± 4.23	2.69 ± 1.27	3.68 ± 2.57	3.21 ± 2.09	NS
MCP-1 (pg/mL)	120.63 ± 36.19	94.98 ± 51.76	107.81 ± 44.92	120.97 ± 44.50	124.60 ± 57.67	122.91 ± 51.10	NS
Post-Study Candidate Gut-dysfunction Markers							
LPS (EU/mL)	0.07 ± 0.047	0.08 ± 0.057	0.07 ± 0.05	0.07 ± 0.02	0.06 ± 0.03	0.06 ± 0.03	NS
LBP (ng/mL)	2201.25 ± 2823.20	1630.45 ± 2287.74	1896.82 ± 2473.80	2484.89 ± 3406.92	1917.26 ± 2334.62	2191.92 0 ± 2867.93	NS
+sCD14 (×10 pg/mL)	6318.33 ± 1845.33	7393.96 ± 1103.19	6892.00 ± 1541.57	6851.46 ± 1855.94	7739.69 ± 1778.08	7295.57 ± 1843.93	NS
Post-Study Liver Cell Death Markers							
K18M65 (IU/L) ^b^	239.57 ± 106.56	373.50 ± 483.61	311.00 ± 355.79	357.25 ± 129.83	726.00 ± 1678.11	541.62 ± 1185.69	NS
K18M30 (IU/L)	508.47 ± 794.91	247.45 ± 154.50	369.26 ± 548.55	239.59 ± 76.14	246.68 ± 148.57	243.02 ± 114.95	NS
M65:M30 ^b^	0.98 ± 0.66	1.16 ± 0.80	1.08 ± 0.72	1.56 ± 0.52	1.35 ± 0.56	1.46 ± 0.54	NS

ALT: serum alanine aminotransferase, AST: serum aspartate aminotransferase, AST:ALT—ratio of AST by ALT, CONUT: Controlling Nutritional Status Test (unit: numerical), WBC: white blood cells count, AMC: absolute monocyte count, ANC: absolute neutrophil count, Il1β: interleukin 1 beta, IL-6: interleukin 6, TNFα: tumor-like necrotic factor alpha, LPS: lipopolysaccharide, LBP: LPS binding protein, sCD14: soluble cell of differentiation type 14, K18M65: soluble CK18, K18M30: caspase-cleaved fragment of CK18, M65:M30—ratio of K18M65 by K18M30. ^a^ Statistically significant difference between the hypomagnesemia sub-groups of the two groups. ^b^ Statistically significant difference between the sub-groups exhibiting normal magnesium levels of the two groups.

## Data Availability

Data can be provided by contacting the corresponding author on appropriate requests.

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
