# Peer review of "Association of Hypomagnesemia and Liver Injury, Role of Gut-Barrier Dysfunction and Inflammation: Efficacy of Abstinence, and 2-Week Medical Management in Alcohol Use Disorder Patients"

_ijms, 2022, doi:10.3390/ijms231911332_

Round 1

Reviewer 1 Report

The authors proposed that serum magnesium level as well as level of inflammatory cytokines, e.g., LBP, TNF alpha, can be the specific markers for hepatocellular apoptosis and necrosis in early-stage alcoholic liver disease (ALD). This manuscript will be of interest to scientist studying early detection of diseases that are believed to be asymptomatic in the early stages. Please find below just few requests to improve the quality of the manuscript.

Essential revisions:

·      The Introduction section should have backgrounds regarding (1) effects of alcohol intake on immunity and (2) age and gender differences in liver injury. To do this, the aim and significance of this study will be clearer for the readers.

·       Do you think that patients in Gr. 1 are in early-stage of ALD? If so, it is important to show association of serum magnesium level vs. HDD90, AST, and ALT at baseline of Gr. 1, as you showed Figure 3 and 4 for Gr. 2. If not, please discuss ALD markers of Gr.1, since most males in Gr.1 showed normal magnesium level in Fig.1b.

·       Additional analysis to provide detailed information regarding serum chemistry and cytokine assay in Table 1 and 2 would be helpful, i.e., please include information dividing by gender.

Minor revisions:

Line 200 (Page 8): In Discussion section, please use parenthetical references for related Figures and Tables at the end of a sentence. While I could understand looking back Tables and Figures, it took longer than I thought it would be.

There are a number of minor concerns regarding the text that should be corrected. Careful proofreading will help. I specify a few:

Line 22 (Page 1): Please check space before unit, e.g., >40 U/L, >41U/L.

Line 30 (Page 1): Please use consistent abbreviation, e.g., “Gr.2” in line 22, “gr.2” in line 30,  “Gr. 2” in line 345.

Line 32 (Page 1): Please indicate unit of magnesium level.

Line 33 (Page 1): Please define [K18(M30)] and [K18(M650)] in this section or the Introduction.

Line 325 in page 11: Please use consistent abbreviation, “K18  M30” or “CK18  M30”.

Author Response

Essential revisions:

  •   The Introduction section should have backgrounds regarding (1) effects of alcohol intake on immunity and (2) age and gender differences in liver injury. To do this, the aim and significance of this study will be clearer for the readers.

Response: We agree with the reviewer’s comment. (1) We have added the effects of alcohol and immunity in context of liver and brain. (2) We have also addressed the modifying effects of age and sex differences in the introduction.

  • Do you think that patients in Gr. 1 are in early-stage of ALD? If so, it is important to show association of serum magnesium level vs. HDD90, AST, and ALT at baseline of Gr. 1, as you showed Figure 3 and 4 for Gr. 2. If not, please discuss ALD markers of Gr.1, since most males in Gr.1 showed normal magnesium level in Fig.1b.

Response: We agree with the reviewer and have addressed the issue in the discussion section. Our investigation aims to identify the candidate drinking biomarkers, that are contributing to the liver injury. In Group 1, where there is no liver injury, thus drinking markers also did not tally adherently with the liver injury markers.

However, we pursued the suggestion of the reviewer and have analyzed the data differently as in Fig.1 b and c (New Figures). Certainly, there is a difference in the magnesium level by males and females (Fig.1b). However, this difference did not exist when we analyzed the data with sex as the primary factor instead of ALT (where sex was the primary factor and ALT was sub-factor with males and females separately), data not shown. Thus, we continued with our original aims to review the data by the factor of ALT and sub-grouping by magnesium and not by sex-differences.

  • Additional analysis to provide detailed information regarding serum chemistry and cytokine assay in Table 1 and 2 would be helpful, i.e., please include information dividing by gender.

Response: We agree with the reviewer and have added some additional details in the Methods section. We have provided the numbers in Table 1 for the females. It would have been desirable to have findings based on the sex-differences. Unfortunately, the numbers in of females in each sub-groups are in low number (that will not yield any statistical usefulness) due to which making a separation as a sub-sub-group within a sub-group will not support any corroborating statistical power. We have added this as a limitation in the discussion section. As also noted, we have conducted sex-difference analyses on such statistical models that have 6 or more subjects’ inventory.

 Minor revisions:

Line 200 (Page 8): In Discussion section, please use parenthetical references for related Figures and Tables at the end of a sentence. While I could understand looking back Tables and Figures, it took longer than I thought it would be.

Response: There is no question of not agreeing to the comment, however, we addressed the ability

There are a number of minor concerns regarding the text that should be corrected. Careful proofreading will help. I specify a few:

Line 22 (Page 1): Please check space before unit, e.g., >40 U/L, >41U/L.

Response: We agree with the reviewer and have corrected as per the comment.

Line 30 (Page 1): Please use consistent abbreviation, e.g., “Gr.2” in line 22, “gr.2” in line 30,  “Gr. 2” in line 345.

Response: We agree with the reviewer and have Used Gr.1 and Gr.2 consistently in the revised version now.

Line 32 (Page 1): Please indicate unit of magnesium level.

Response: We have added the unit for magnesium level.

Line 33 (Page 1): Please define [K18(M30)] and [K18(M650)] in this section or the Introduction.

Response: We agree with the reviewer and have added the explanation on K18s in the introduction section.

Line 325 in page 11: Please use consistent abbreviation, “K18  M30” or “CK18  M30”.

Response: We agree with the reviewer and have now used K18M30 and K18M65 throughout the revised manuscript.

Reviewer 2 Report

The manuscript by Winrich et al., evaluated blood magnesium levels in early alcoholic liver disease (ALD), gut barrier dysfunction, and inflammation in AUD patients, and the effectiveness of 2-week abstinence and medicinal therapy to ameliorate hypomagnesemia. Despite the fact that the article presents significant data. Extensive grammatical errors must also be resolved.

General Comments:

Q1. Authors have used statistical analysis of liver disease in AUD patients, although it is unclear how the author characterizes whether or not certain individuals already have a liver injury.

Q2. Except for the legend explanation, where is the citation for Figures 1a,1b,1c, and description?

Q3. All of the figures' resolutions should need to be changed.

Q4. The abstract definitely needs some editing since it's so long.

Q5. The discussion should be revised, and a recent reference should be included. This study's limitations should be mentioned.

Minor comments:

L22:  Gr.1 and Gr.2.  speak it out at the first time.

L26: Demographics and drinking

L26-27: It's an odd statement that needs rewriting.

L28: significant or a significant, clarify?

L41: necrotic type or necrotic types

L47: disease or diseases

L48, cirrhosis

L49: Studies have found: how many studies? Re-write sentence, be specific, and add the recent reference

L50: paucity or lack

L63: higher intake of magnesium or higher intake of magnesium

L68: proinflammatory or pro-inflammatory be specific through the text.

L100: Serum magnesium or The serum magnesium

L103: level or levels., clarify through the text.

L123: females of this study or females in this study

L124: , magnesium levels inversely associated or , magnesium levels were inversely associated

L131: reference normal or normal reference

L206: all the three or all three

L49:  significant effect or A significant effect

L208: Notably, females were found to have a greater adverse response of heavy drinking which corresponded to the lower magnesium level. Or Notably, females were found to have a greater adverse response to heavy drinking which corresponded to a lower magnesium level.

L240: high however correct it

Author Response

General Comments:

Q1. Authors have used statistical analysis of liver disease in AUD patients, although it is unclear how the author characterizes whether or not certain individuals already have a liver injury.

Response: The methods section has the explanation available. In this journal, the format of the manuscript limits us to write the methods after discussion. Thus, abiding with the journal requirements, the explanation are limited to the methods section.

Q2. Except for the legend explanation, where is the citation for Figures 1a,1b,1c, and description?

Response: We agree with the reviewer and citations for the figures are now noted in the results section.

Q3. All of the figures' resolutions should need to be changed.

Response: We have improved the resolution of the manuscript.

Q4. The abstract definitely needs some editing since it's so long.

Response: We agree with the reviewer and have edited the abstract.

Q5. The discussion should be revised, and a recent reference should be included. This study's limitations should be mentioned.

Response: We agree with the reviewer. We have revised the discussion section. We have also provided some additional references in the introduction and discussion section. There is a full paragraph on the limitations of this study in the discussion section that was mentioned in the original submission. We have added an opening statement to clearly identify this paragraph.

Minor comments:

L22:  Gr.1 and Gr.2.  speak it out at the first time.

Response: we have added the full forms in the abstract and in the results section, where these terms appeared first.

L26: Demographics and drinking

Response: we agree with the reviewer and have added “and” between the two words.

L26-27: It's an odd statement that needs rewriting.

Response: we agree with the reviewer and have revised the statement.

L28: significant or a significant, clarify?

Response: We agree with the reviewer and although we removed that statement as it was not the highlight of the findings that we added some new data that was more exciting. However, in other applicable statements, we made suggested changes (for example, the first statement [Female …] in the results of the abstract).

L41: necrotic type or necrotic types

Response: we agree with the reviewer and have made the change.

L47: disease or diseases

Response: We agree with the reviewer and have made change.

L48, cirrhosis

Response: we have added space between fibrosis/cirrhosis in the revised version.

L49: Studies have found: how many studies? Re-write sentence, be specific, and add the recent reference.

Response: We agree with the reviewer and have edited the sentence and added new references.

L50: paucity or lack

Response: We agree with the reviewer that the word is not the best fitting word and have changed it.

L63: higher intake of magnesium or higher intake of magnesium

Response: we agree with the reviewer and have made appropriate change.

L68: proinflammatory or pro-inflammatory be specific through the text.

Response: We agree with the reviewer and have now used pro-inflammatory” in the manuscript.

L100: Serum magnesium or The serum magnesium

Response: We agree with the reviewer and have made appropriate change (though it was on line 87 and we changed it at that location).

L103: level or levels., clarify through the text.

Response: We agree with the reviewer and have made changes throughout the manuscript using only “Level”.

L123: females of this study or females in this study

Response: We have removed this statement; thus, this comment is not accommodated. We found that the K18 data was more novel and exciting, thus we use the K18 data in the new figures and corresponding statement in the results section in the revised version.

L124: , magnesium levels inversely associated or , magnesium levels were inversely associated

Response: We have removed this statement; thus, this comment is not accommodated. We found that the K18 data was more novel and exciting, thus we use the K18 data in the new figures and corresponding statement in the results section in the revised version.

L131: reference normal or normal reference

Response: we agree with the reviewer and have change to “normal reference”.

L206: all the three or all three

Response: We agree with the reviewer and have removed “the”. 

L49:  significant effect or A significant effect

Response: we agree with the reviewer and have added “a” before “significant effect” (Line 115 of the original file).

L208: Notably, females were found to have a greater adverse response of heavy drinking which corresponded to the lower magnesium level. Or Notably, females were found to have a greater adverse response to heavy drinking which corresponded to a lower magnesium level.

Response: We agree with the reviewer and with minimal change address the sentence that was suggested to use.

L240: high however correct it

Response: we agree with the reviewer and have corrected the error.

Reviewer 3 Report

This is a retrospective study investigating the relationship between hypomagnesemia and liver injury in patients with alcohol use disorder. I could not understand well why the authors focused on hypomagnesemia. Why was the division to group 1 and group 2 necessary? The conclusion is not well described in the abstract.

Author Response

This is a retrospective study investigating the relationship between hypomagnesemia and liver injury in patients with alcohol use disorder. I could not understand well why the authors focused on hypomagnesemia. Why was the division to group 1 and group 2 necessary? The conclusion is not well described in the abstract.

Response: This is not a retrospective study. This is a clinical baseline to post-treatment study with a pilot level of patient participation.

Hypomagnesemia worsens significantly with the progression of ALD. We have mentioned about such information in the introduction and discussion section already.

The two groups drink similarly, but the Gr.2 gets early-stage ALD. Thus , we identify the clues that are involved in the onset of the disease and what measures of medical management may be helpful.

The research laboratory testing were performed to address the endogenous variables that could go along with the clinical laboratory and clinical presentation of the patients with may have hypomagnesemia. This is a continuation of our ongoing focus on the clinical trials are being performed and this study focuses on the early-stage of ALD. Our several studies have focused also on advanced state of ALD and this study identifies the indicators of early-stage ALD, how to characterize the features of liver cell death and liver injury in context of the treatment program. Previously, we have published hypomagnesemia in a relatively large population with similar clinical characteristics (Vatsalya, 2020); present study attempts to explore the role of gut-dysfunction, proinflammatory immune status and the novel cell death markers of the liver. This study also extends the understanding of the medical management that is used and its efficacy; that provides the knowledge of the direction of medical management that could have a course in the advanced form of the ALD (We are actively working on the hypomagnesemia studies on alcoholic hepatitis and a presentation on it is being presented at AASLD 2022). We have also updated the abstract.

Round 2

Reviewer 1 Report

This manuscript has improved a lot. The analysis and conclusion are convincing and suitable for publication in present form.

Reviewer 3 Report

The authors have revised the manuscript appropriately.